# Relationship between Very Fine Root Distribution and Soil Water Content in Pre- and Post-Harvest Areas of Two Coniferous Tree Species

**Moein Farahnak** [1,2], **Keiji Mitsuyasu** [1], **Takuo Hishi** [2], **Ayumi Katayama** [2], **Masaaki Chiwa** [2], **Seonghun Jeong** [1,2], **Kyoichi Otsuki** [2], **Seyed Mohammad Moein Sadeghi** [3] **and Atsushi Kume** [4,*]

1   Graduate School of Bioresources and Bioenvironmental Science, Kyushu University, Fukuoka 8190395, Japan; moein.farahnak@gmail.com (M.F.); keiji.m@outlook.com (K.M.); nowwap@gmail.com (S.J.)
2   Kyushu University Forest, Kyushu University, Fukuoka 8112415, Japan; hishi@forest.kyushu-u.ac.jp (T.H.); ayumi@forest.kyushu-u.ac.jp (A.K.); mchiwa@forest.kyushu-u.ac.jp (M.C.); otsuki@forest.kyushu-u.ac.jp (K.O.)
3   Technical Bureau of Forestry and Plantation, Natural Resources and Watershed Management Office of West Azerbaijan Province, Urmia 5719975898, Iran; moeinsadeghi@ut.ac.ir
4   Department of Agro-Environmental Science, Kyushu University, Fukuoka 8190395, Japan
*   Correspondence: akume@agr.kyushu-u.ac.jp; Tel.: +81-92-802-4674

**Abstract:** Tree root system development alters forest soil properties, and differences in root diameter frequency and root length per soil volume reflect differences in root system function. In this study, the relationship between vertical distribution of very fine root and soil water content was investigated in intact tree and cut tree areas. The vertical distribution of root density with different diameter classes (very fine <0.5 mm and fine 0.5–2.0 mm) and soil water content were examined along a slope with two coniferous tree species, *Cryptomeria japonica* (L.f.) D. Don and *Chamaecyparis obtusa* (Siebold et Zucc.) Endl. The root biomass and length density of very fine roots at soil depth of 0–5 cm were higher in the *Ch. obtusa* intact tree plot than in the *Cr. japonica* intact plot. Tree cutting caused a reduction in the biomass and length of very fine roots at 0–5 cm soil depth, and an increment in soil water content at 5–30 cm soil depth of the *Ch. obtusa* cut tree plot one year after cutting. However, very fine root density of the *Cr. japonica* intact tree plot was quite low and the soil water content in post-harvest areas did not change. The increase in soil water content at 5–30 cm soil depth of the *Ch. obtusa* cut tree plot could be caused by the decrease in very fine roots at 0–5 cm soil depth. These results suggest that the distribution of soil water content was changed after tree cutting of *Ch. obtusa* by the channels generated by the decay of very fine roots. It was also shown that differences in root system characteristics among different tree species affect soil water properties after cutting.

**Keywords:** *Cryptomeria japonica*; *Chamaecyparis obtusa*; forest management; post-harvest; very fine root; soil water content

## 1. Introduction

Tree roots are vital components of forest ecosystems because they connect aboveground parts of a tree to the soil. Coarse roots are responsible for anchoring the tree into the soil, while fine roots explore the soil to take up water and nutrients [1]. The boundary between coarse and fine roots is usually defined as a diameter of 2 mm (e.g., [2,3]). However, in some studies, roots with a diameter of <0.5 mm are considered very fine or finest roots (e.g., [4,5]). In general, the vertical and horizontal distributions of both very fine and fine roots are related to endogenous [6] and exogenous (environmental) factors [7–9]

in soil. Soil water content is one of the exogenous factors that link to the distribution of very fine and fine roots of a standing tree [7–9]. The availability of nutrients also integrates with soil water content to affect the distribution of roots in the soil [10,11]. Previous studies have shown that the distribution of fine roots is correlated with soil water content [4,12,13]. In humid and temperate regions, shallow root systems tend to be well developed, resulting in uneven distribution of soil moisture and nutrients [14].

In Japan, 67% of the land is covered with forest areas [15]. Most forest areas are in the mountains and play an essential role in the water cycle. *Cryptomeria japonica* (L.f.) D. Don and *Chamaecyparis obtusa* (Siebold et Zucc.) Endl are two common Japanese coniferous tree species planted broadly in Japan [15]. The root system differs for these two species, i.e., *Cr. japonica* develops a deep root system, and *Ch. obtusa* develops a shallow root system [16–18]. Despite similar rainfall partitioning in the aboveground part of these two tree species [19–22], underground soil water movement varies because of different forest floor conditions; *Cr. japonica* forest develops a thick litter layer, and the soil surface shows high soil hydraulic conductivity. In contrast, *Ch. obtusa* forest has a thin litter layer, and the soil surface shows high soil water repellency [18,23]. Wide areas of *Cr. japonica* and *Ch. obtusa* plantations have been abandoned/unmanaged in the last decades, which has caused changes in the water cycle of mountain areas [24,25]. Policymakers and forest managers have been considering different methods to manage these planted areas. One of the proposed methods is the conversion from coniferous to broadleaved forests, while the other main method, which is broadly accepted and operated in Japan, is either thinning or clear-cutting. Previous studies have evaluated the consequences of forest harvesting (either thinning or clear-cutting) on the ecohydrological processes of post-harvest areas [19,20,26–28]. These studies evaluated changes in aboveground and belowground ecohydrological processes in post-harvest areas. However, uncertainty remained about the impact of very fine roots on soil hydrology alterations. Therefore, the relationship between soil water content and the root system in pre-harvest and post-harvest areas is an indispensable factor to understand and evaluate forest ecohydrological management.

The relationship between very fine root and fine root characteristics, such as biomass, length, and specific root length (SRL) with soil depth, has been elucidated in *Cr. japonica* and *Ch. obtusa* plantations (regarding deep and shallow root systems, e.g., [29,30]). However, the relationships among root characteristics, especially very fine root, are still unclear with soil water content because of their decomposition after tree harvesting.

This study aimed to investigate whether differences in the properties of the root systems between the two species, *Cr. japonica* and *Ch. Obtuse*, could affect the distribution of soil water content in the post-harvest areas. We surveyed the vertical changes in the biomass and length of root density (very fine root and fine root) in pre- and post-harvest areas and analyzed their relationships with soil water content distribution.

## 2. Materials and Methods

### 2.1. Study Site

This study was conducted at the experimental plantation of Kyushu University in the south of Japan (33°38′ N, 130°31′ E, ca. 130 m a.s.l). *Cr. japonica* and *Ch. obtusa* were planted in 1934 (0.5 ha) and 1961 (5.2 ha), respectively. The plantations both have a density of 1600 trees ha$^{-1}$. The forest operation was conducted in a part of this site (1.14 ha), in early 2016, thinning of *Cr. japonica* and clear-cutting of *Ch. obtusa*. The current study was conducted in 2017, one year after logging. The mean annual air temperature and precipitation (2007–2016) of the nearest meteorological station located 15 km southwest of the study site were 17 °C and 1700 mm, respectively (Hakata Meteorological Station, 33°36′ N, 130°25′ E, 3 m a.s.l). The understory was scarce in both intact tree plots, while annual herbs started growing in both cut tree plots. The forest floor had different conditions between *Cr. japonica* and *Ch. obtusa* plots. Litter layer depth (cm) and litter dry biomass (g m$^{-2}$) were 2.85 cm and 1530 g m$^{-2}$ in *Cr. japonica* intact tree plots; 2.53 cm and 363 g m$^{-2}$ in *Ch. obtusa* intact tree plots; 3.45 cm and

885 g m$^{-2}$ in *Cr. japonica* cut tree plots; and 0.0 cm and 0.0 g m$^{-2}$ in *Ch. obtusa* cut tree plots, respectively. The soil type is classified as brown forest soil [31], which is equivalent to Cambisol [32] that originated from serpentine bedrock. The soil properties of the studied plots are given in Table 1 (see [18] for a detailed description).

*2.2. Sampling Design and Soil Collection*

Four study plots were selected along a southwest slope face (~30°) that included the following: (1) *Cr. japonica* intact tree, (2) *Ch. obtusa* intact tree, (3) *Cr. japonica* cut tree, and (4) *Ch. obtusa* cut tree. Mean tree height and diameter at breast height were 17.6 m and 30 cm in the *Cr. japonica* intact tree plot and 21.2 m and 31 cm for the *Ch. obtusa* intact tree plot, respectively. We selected five individual trees and stumps as replicates in each plot. Soil samples were collected from the upslope and downslope of each individual tree and stump at different distances (0.5 m and 1.0 m) and soil depths (0–5, 5–10, and 10–30 cm). These two slope positions (i.e., upslope and downslope) were selected because they were the areas most affected in terms of soil properties by trees and stumps in the slope areas [18]. However, we pooled the data of upslope and downslope areas and different distances as one sample at each soil depth in each plot. This is because the data did not show significant differences for different slope positions (i.e., upslope and downslope) and distances (i.e., 0.5 m and 1.0 m) for the measured parameters (i.e., root characteristics and soil water content). Therefore, 20 soil samples were collected from each soil depth of each plot (a total of 60 samples in each plot). Samples for either roots or soil experiments were collected from May to June for intact tree plots and from October to November for cut tree plots (Supplementary Materials Figure S1). The collection of samples was conducted at least 48 h after the last rain event (Supplementary Materials Figure S1).

*2.3. Experiments*

2.3.1. Soil Experiments

Soil near-saturated hydraulic conductivity of the soil surface was measured by tension mini disk infiltrometer (Decagon Devices, Inc., Pullman, WA, USA) A steel cylinder (volume 100 cm$^3$) was used to collect soil samples to measure soil dry bulk density and soil volumetric water content ($\theta_v$) at each soil depth. Soil particle density was measured using the pycnometer method to calculate the soil core volume (*Cr. japonica* plots, 2.55 g cm$^{-3}$ and *Ch. obtusa* plots, 2.52 g cm$^{-3}$). Soil texture and soil organic matter (SOM) content were measured by the hydrometer and loss on ignition methods, respectively [33]. Soil calcium ion (Ca$^{2+}$) was measured with atomic absorption spectroscopy (AAS, AA-7000, Shimadzu, Kyoto, Japan). Soil pH (KCl) was measured by the glass electrode method (soil/water = 1:2.5) using a pH meter (D-54, Horiba Ltd., Kyoto, Japan). The potential soil water repellency intensity was measured with the molarity of an ethanol droplet test [23]. For more soil experimental methodology, see [18,23].

**Table 1.** Soil properties in the studied plots.

| Tree Species | *Cryptomeria Japonica* | | | | | | *Chamaecyparis Obtusa* | | | | | |
|---|---|---|---|---|---|---|---|---|---|---|---|---|
| Plots | Intact Tree | | | Cut Tree | | | Intact Tree | | | Cut Tree | | |
| Soil depth (cm) | 0–5 | 5–10 | 10–30 | 0–5 | 5–10 | 10–30 | 0–5 | 5–10 | 10–30 | 0–5 | 5–10 | 10–30 |
| Sand (%) | 63.13 | 61.11 | 47.56 | 70.03 | 60.92 | 55.78 | 75.14 | 59.54 | 61.47 | 73.08 | 65.61 | 55.96 |
| Clay (%) | 10.94 | 13.26 | 25.53 | 9.72 | 14.41 | 15.72 | 9.28 | 15.17 | 12.60 | 11.78 | 13.00 | 15.92 |
| Soil bulk density | 1.06 | 1.09 | 1.23 | 0.87 | 1.05 | 1.13 | 0.91 | 1.04 | 1.18 | 0.86 | 1.02 | 1.15 |
| Soil organic matter | 12.99 | 9.21 | 8.11 | 11.66 | 8.86 | 7.29 | 13.07 | 8.59 | 7.42 | 14.22 | 10.21 | 8.00 |
| $Ca^{2+}$ | 11.08 | 5.92 | 3.07 | 14.51 | 9.61 | 5.97 | 2.87 | 2.40 | 3.20 | 5.35 | 5.19 | 4.96 |
| Soil pH (KCl) | 4.03 | 3.92 | 3.80 | 4.30 | 4.08 | 3.87 | 3.47 | 3.59 | 3.74 | 3.71 | 3.82 | 3.95 |
| Soil water repellency * | 1.80 | 0.0 | 0.0 | 3.05 | 0.0 | 0.0 | 13.10 | 3.60 | 0.0 | 11.65 | 3.50 | 0.0 |
| Soil hydraulic conductivity | 9.59 | – | – | 13.51 | – | – | 2.17 | – | – | 2.75 | – | – |

The sample number at each soil depth was 20, * the number of samples for soil water repellency (ethanol %) at each depth was 10 (collected 0.5 m from individual trees and stumps). Soil bulk density (g cm$^{-3}$) and soil organic matter (%), Ca$^{+2}$ (cmol(+) kg$^{-1}$). The soil near-saturated hydraulic conductivity (k, mm h$^{-1}$). The data for this table were retrieved from [18,23].

### 2.3.2. Root Experiments

A metal auger with a removable inner plastic tube (5 cm in diameter and 30 cm in depth) was used to collect root samples. The metal auger was hammered vertically into the soil in different slope positions (i.e., upslope and downslope) and distances (i.e., 0.5 m and 1.0 m) from individual trees and stumps to collect root samples. Collected plastic tubes were transferred to the laboratory within 12 h. The plastic tubes were cut at three soil depths, i.e., 0–5, 5–10, and 10–30 cm, and roots were removed from soils by washing in tap water [17]. Root samples were divided into the following four subsamples: very fine root (diameter, x < 0.5 mm), fine root (diameter, 0.5 < x < 2.0 mm), other understory or vegetation roots (other, diameter, x < 2.0 mm), and coarse root (diameter, x > 2.0 mm, excluded from analysis). *Cr. japonica*, *Ch. obtusa*, and other plant roots were distinguished based on their color and texture [29,30,34,35]. However, we did not divide roots as live or dead roots. The root samples were scanned (600 dpi resolution, grayscale), and then put inside a paper envelope in the oven for 48 h at 50 °C to determine their dry weight of biomass. The scanned images were used to measure root length using ImageJ software (Ver. 1.8.0–112, [36]). Specific root length (SRL, m g$^{-1}$) was calculated by dividing root length (m) by root biomass (g). Root biomass and length density were calculated using the following equations:

$$\text{Root biomass density}\left(g/m^3\right) = \frac{\text{Biomass (g)}}{\pi r^2 h \ (m^3)} \tag{1}$$

$$\text{Root length density}\left(cm/cm^3\right) = \frac{\text{length (cm)}}{\pi r^2 h \ (cm^3)} \tag{2}$$

where r and h are the tube radius and corresponding soil depth, respectively.

### 2.4. Statistical Analysis

Pooled data of different slope positions and distances at each soil depth (20 samples for each soil depth) were expressed as mean ± standard error (SE). The analysis of variance (ANOVA) and the Tukey's honestly significant difference (HSD) test (at a 5% significance level) were used to detect the differences in soil volumetric water content and root characteristics (biomass, length, and SRL) of different categories (very fine, fine, other, and total roots) in different tree species (*Cr. japonica* and *Ch. obtusa*), intact tree and cut tree plots, and for different soil depths (0–5, 5–10, and 10–30 cm). Pearson's correlation (*r*) analysis was performed between $\theta_v$ and root characteristics (biomass, length and SRL of very fine, fine, other, and total root density) at 5% significance level. For this correlation (*r*), we pooled data for each soil depth (i.e., 0–5, 5–10, 10–30 cm) in all four plots and the entire soil profile (i.e., 0–30 cm) of the studied slope for $\theta_v$ and root characteristics. These correlations were conducted to understand the horizontal (i.e., pooled data of each soil depth) and vertical (i.e., pooled data of the entire soil profile) relationship between soil water content and root characteristics. All statistical analyses were performed using R software ver. 3.4.2 (R Development Core Team, Vienna, Austria).

## 3. Results

### 3.1. Soil Volumetric Water Content

$\theta_v$ was significantly higher in the entire soil profile of *Cr. japonica* plots than that of *Ch. obtusa* (Figure 1). $\theta_v$ was similarly distributed in the soil of *Cr. japonica* intact and cut tree plots. However, there were significant differences between $\theta_v$ of *Ch. obtusa* intact and cut tree plots at 5–30 cm (Figure 1).

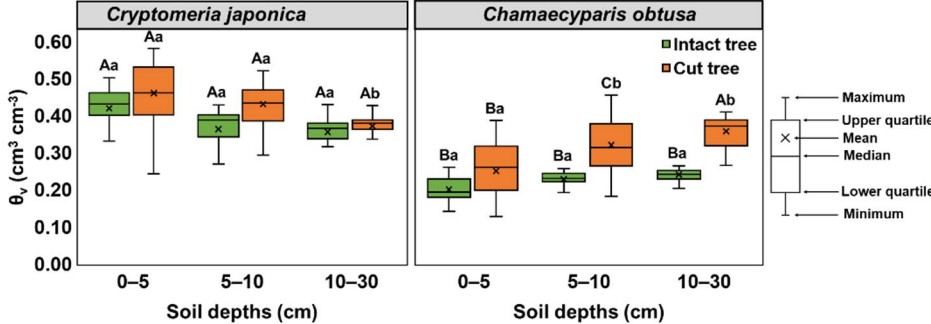

**Figure 1.** Soil volumetric water content ($\theta_v$, cm$^3$ cm$^{-3}$) at different soil depths of *Cryptomeria japonica* and *Chamaecyparis obtusa* intact tree and cut tree plots. Capital and small letters denote significant differences between plots and soil depths, respectively (*n* = 20) (data were retrieved from [18]).

*3.2. Root Characteristics*

3.2.1. Biomass Density

The total root biomass density was significantly larger at 0–5 cm soil depth for the *Ch. obtusa* intact tree plot than that of the *Cr. japonica* intact tree and *Ch. obtusa* cut tree plots (Figure 2a). The total root biomass density declined significantly with increasing soil depth to 5–10 and 10–30 cm soil depth of the *Ch. obtusa* intact tree plot (Figure 2a). Very fine root length density accounted for a large percentage of the total root in the *Ch. obtusa* intact tree plot at 0–5 cm (Figure 2b). The differences in total root biomass density between the two tree species and intact and cut tree plots were mostly related to the allocation of very fine root biomass density at 0–5 cm soil depth (Figure 2a,b). In contrast, fine root and other root had similar biomass between the studied plots at 0–30 cm (Figure 2a).

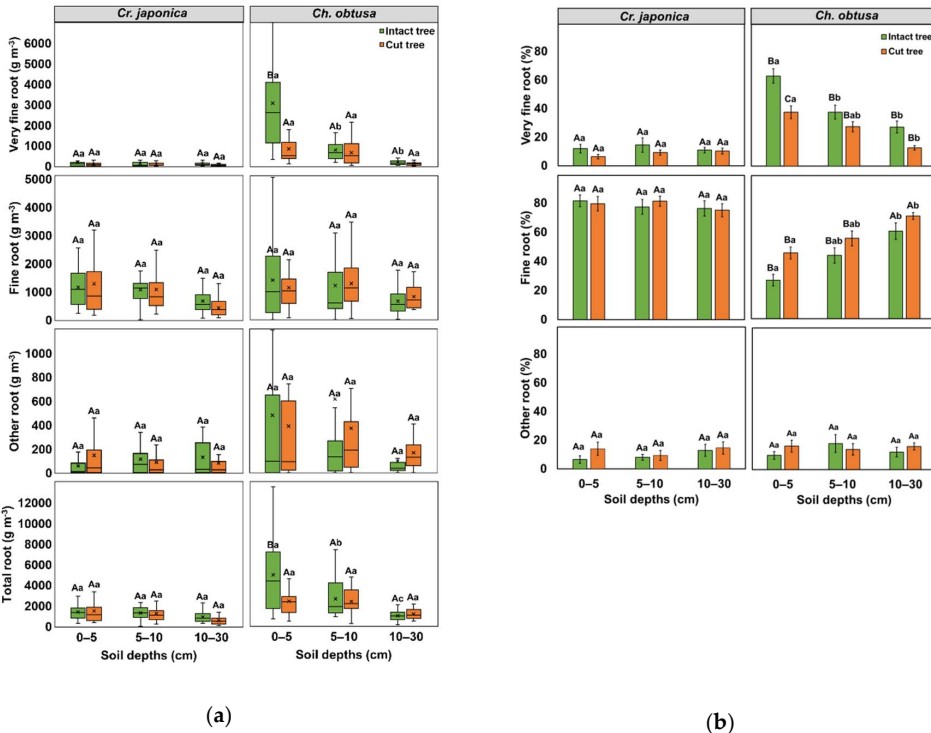

(**a**)                                            (**b**)

**Figure 2.** Root biomass density (g m$^{-3}$) at different soil depths of *Cryptomeria japonica* and *Chamaecyparis obtusa* intact tree and cut tree plots. (**a**) Root (very fine, fine, other, and total) biomass density; (**b**) Root (very fine, fine, and other) biomass density percentage to total root biomass density. Capital and small letters denote significant differences between plots and different soil depths, respectively (*n* = 20). Root biomass density data were retrieved from [18].

### 3.2.2. Length Density

In the *Ch. obtusa* intact tree plot, the root length density at 0–5 cm soil depth was higher than that of *Cr. japonica* intact tree and *Ch. obtusa* cut tree plots (Figure 3a). The total root length density significantly declined with soil depth in the *Ch. obtusa* intact tree plot (Figure 3a). Very fine root length density also accounted for a large percentage of the total root in the *Ch. obtusa* intact tree plot at 0–5 cm (Figure 3b). The differences in root length density for intact tree plots of both coniferous species were related to the very fine root length, with high and low density (percentage of total root length density) in *Ch. obtusa* and *Cr. japonica* intact tree plots, respectively, at 0–5 cm soil depth (Figure 3a,b). In the *Ch. obtusa* cut tree plot, the other root length density was higher than the intact tree plot at 0–30 cm (Figure 3a).

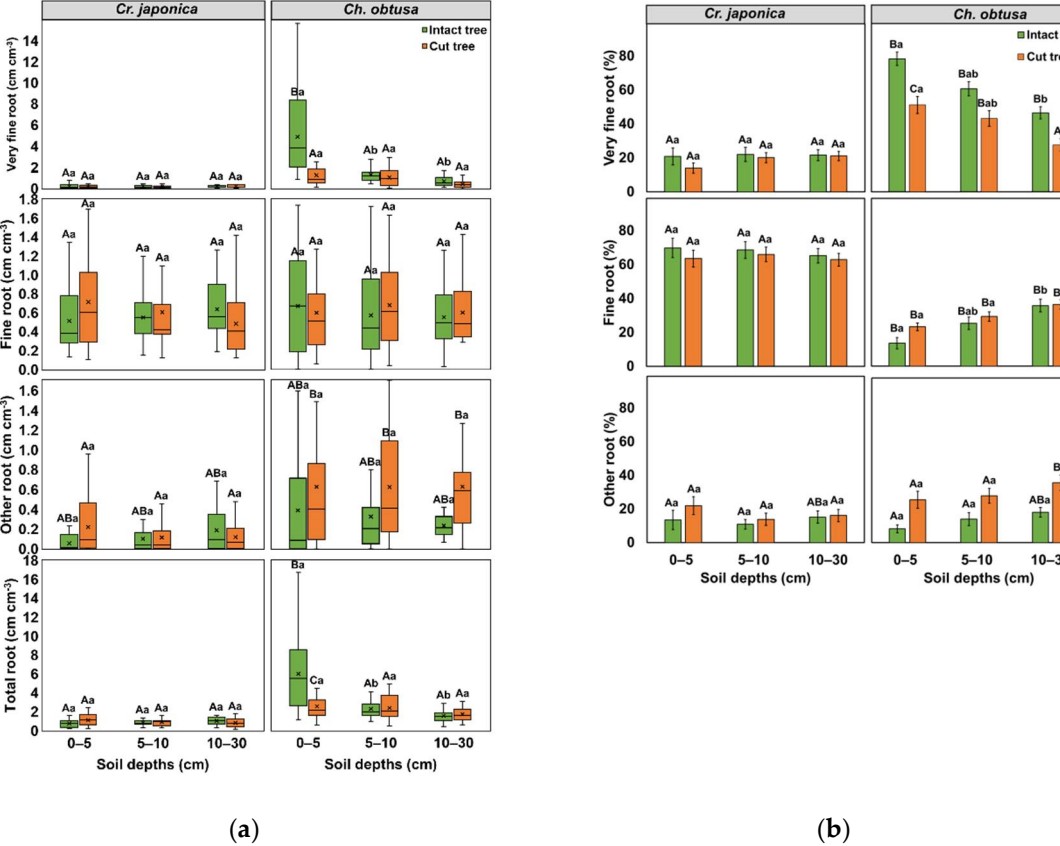

(**a**)  (**b**)

**Figure 3.** Root length density (cm cm$^{-3}$) for different soil depths of *Cryptomeria japonica* and *Chamaecyparis obtusa* intact tree and cut tree plots. (**a**) Root (very fine, fine, other, and total) length density; (**b**) Root (very fine, fine, and other) length density percentage to total root length density. Capital and small letters denote significant differences between plots and different soil depths, respectively ($n = 20$).

### 3.2.3. Specific Root Length

The SRL of the very fine root was significantly different at each soil depth of the *Cr. japonica* intact tree plot (Supplementary Materials Figure S2). The SRL of very fine root increased from 0–5 to 5–10 cm, and then decreased to 10–30 cm soil depth of the *Cr. japonica* intact tree plot (Supplementary Materials Figure S2).

### 3.3. Soil Water Content Relationships with Root Characteristics

$\theta_v$ was negatively correlated with both densities of biomass and length of very fine root in all studied plots (Figure 4). The densities of very fine root biomass and length decreased with increasing $\theta_v$ at 0–5 cm soil depth (Figure 4). The correlations were highest at 0–5 cm soil depth and gradually

reduced with increasing soil depth at 5–10 and 10–30 cm (Figure 4). In the entire soil profile (0–30 cm), $\theta_v$ was also negatively correlated with very fine root density (biomass and length) (Figure 4).

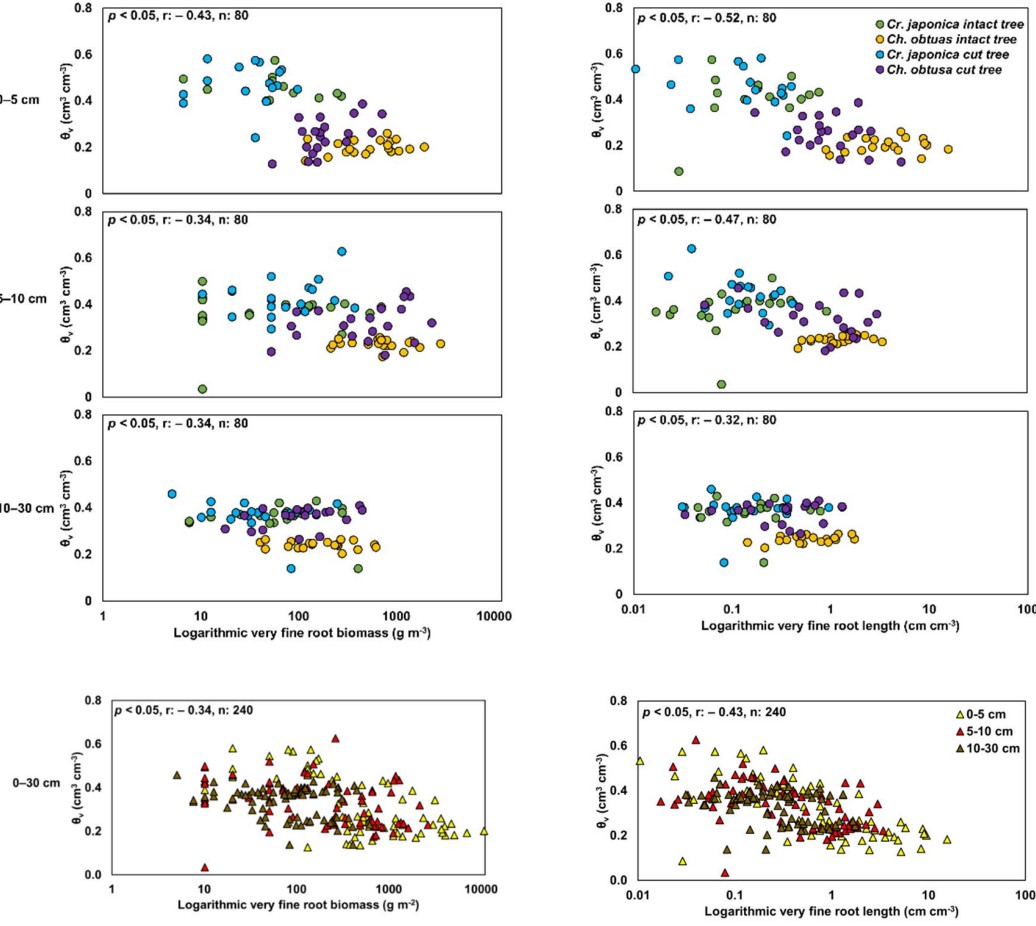

**Figure 4.** Pearson correlation coefficient (*r*) between soil water content ($\theta_v$) and very fine root biomass and root length density for different soil depths of *Cr. japonica* and *Ch. obtusa* intact tree and cut tree plots.

## 4. Discussion

Tree fine roots constitute a dynamic component of forest ecosystems [37,38]. In this study, we investigated the relationship between root distribution and soil water content in pre- and post-harvest areas of two coniferous plantations (Figure 5).

Fine root biomass for both tree species was densely concentrated in the upper soil layer, which was consistent with previous reports [39–41]. Dense fine roots have been reported in other *Cr. japonica* and *Ch. obtusa* plantations in Japan [29,30,42,43]. Changes in fine root biomass are more noticeable in the upper soil layer (0–5 cm) than in the deeper soil layers (5–30 cm) because of the high organic matter and mineral nutrients in the topsoil [41]. The denser very fine root biomass and length in the *Ch. obtusa* intact tree plot as compared with the *Cr. japonica* could be explained in two ways. First, a shortage of $Ca^{2+}$ in *Ch. obtusa* leaves has been mentioned in several studies [44–46]. A deficiency of $Ca^{2+}$ in the soil surface layer has been detected in the soil of the same *Ch. obtusa* plantation investigated in this study (Table 1). This scarcity of $Ca^{2+}$ might influence the root behavior and generate dense very fine roots with a small diameter (<0.5 mm) in shallow soil (0–5 cm) to cover the tree's nutritional demand. Thus, *Ch. obtusa* trees produced very fine roots in shallow soil because their leaf litter contains low available calcium to deliver into the soil surface [45–48]. Miyatani et al. [43] also mentioned that roots with small diameter concentrated in the uppermost soil layer (0–10 cm) to compensate for

the nutrient deficiency of *Ch. obtusa* stands with low acid buffering capacity soil. Gower [48] also showed that fine root biomass might be inversely related to $Ca^{2+}$ availability in the soil. Second, soil in *Ch. obtusa* plantations contained water repellent materials that reduced water infiltration into the soil surface [18,23]. Farahnak et al. [23] showed that instantaneous soil moisture shortage was linked to high soil water repellency around *Ch. obtusa* trees. Therefore, soil water repellency could be another explanation for *Ch. obtusa* trees to invest in an efficient root system (i.e., very fine root) in the soil surface layer to absorb soil water and nutrients, thus, increasing root surface area per biomass. Soil hydrological processes could be affected by soil water repellency through reduced matrix infiltration, fingered flow development, and overall increased runoff generation and soil erosion [49,50]. The combination of these two reasons (lack of $Ca^{2+}$ and high soil water repellency) influenced the concentrated distribution of very fine root and low soil water content at 0–5 cm soil depth of the *Ch. obtusa* intact tree plot. Although Hishi and Takeda [30] suggested that *Ch. obtusa* trees absorb water primarily through roots in the upper layer of soil (organic layer), they also noted that not all fine roots (root diameter <2.0 mm) were capable of water absorption because of physiological changes in root cells during different growth stages. Furthermore, the presence of long very fine roots at 0–5 cm soil depth of the *Ch. obtusa* intact tree plot could reduce infiltration of water into the soil (refer to low soil water content at 0–5 cm soil depth of *Ch. obtusa* intact tree plot, Figure 1). Leung et al. [51] studied the impacts of root characteristics on soil hydraulic conductivity between a grass species (*Cynodon dactylon*) and a tree species (*Schefflera heptaphylla*). They suggested that the presence of long fine roots in soil reduced soil hydraulic conductivity by clogging soil pores [51]. In contrast to the *Ch. obtusa* intact tree plot, soil in the *Cr. japonica* intact tree plot had high $Ca^{2+}$ availability and soil water content that resulted in the low proportion of very fine root at 0–30 cm soil depth. Increased soil water content and $Ca^{2+}$ in the soil surface of the *Cr. japonica* intact tree plot might be related to high soil hydraulic conductivity and the thick litter layer, respectively [18]. Hirano et al. [47] concluded that high acid buffering capacity and $Ca^{2+}$ accumulation induced low allocation of fine root biomass. In contrast, very fine root or thinner root was produced because of low acid buffering capacity and subsequent changes in root cells and morphology in mature *Cr. japonica* plantation.

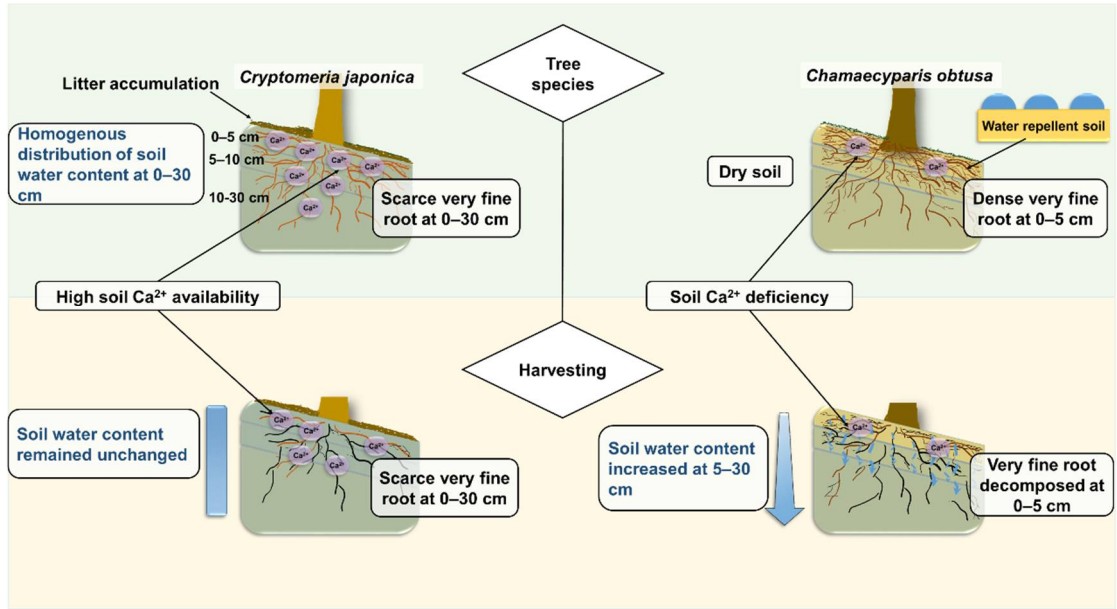

**Figure 5.** Schematic diagram of research results.

Although sampling was undertaken at a different time (Supplementary Materials Figure S1), soil water content showed similar trends with increasing soil depth between the intact tree and cut tree plots of both *Cr. japonica* and *Ch. obtusa*, respectively (Figure 1). This means, soil water content had a

similar decreasing trend between intact and cut tree plots of *Cr. japonica*. In *Ch. obtusa* intact and cut tree plots, a similar increasing trend was observed with increasing soil depth (Figure 1). The higher soil water content of the *Ch. obtusa* cut tree plot than that of the intact tree plot could be linked to removing the aboveground part (canopy) and cessation of water uptake after tree cutting at 5–30 cm.

In the *Ch. obtusa* cut tree plot, the soil surface was exposed to direct solar radiation. Therefore, substantial evaporation and runoff from the soil surface occurred. Additionally, by removing trees, annual plants had opportunities to take up water (Figure 3). These conditions reduced the soil water content of the soil surface (0–5 cm) in the *Ch. obtusa* cut tree plot more than that of the *Ch. obtusa* intact tree plot. The increase in soil water content at 5–30 cm soil depths of the *Ch. obtusa* cut tree plot might be promoted by channels produced by the decomposition of very fine roots in 0–5 cm soil depth. In the *Cr. japonica* cut tree plot, the soil surface remained covered with the remnant tree canopy and litter layer. Therefore, evapotranspiration from the soil surface was limited and contributed to an increase in soil water content of the soil surface layer around individual *Cr. japonica* stumps as compared with intact trees.

## 5. Conclusions

We investigated the relationship between the root distribution and soil water content of trees and stumps of two different coniferous tree species, *Cr. japonica* and *Ch. obtusa*, on a hillslope. Soil water content was negatively correlated with very fine root density (length and biomass). The present study supports a possible association between the very fine roots of *Ch. obtusa* and low soil water content in the soil surface in the intact tree plot. *Ch. obtusa* seems to maintain low soil water content and use water on the soil surface (0–5 cm soil depth). The increase in soil water content at 5–30 cm soil depth may be due to the decomposition of very fine roots at 0–5 cm soil depth in the *Ch. obtusa* cut tree plot. In contrast, *Cr. japonica* forms soil with high soil water content, allowing very fine and fine roots to use soil water at soil depths of 0–30 cm. These findings suggested that soil hydrological properties were affected by very fine root decomposition in the *Ch. obtusa* post-harvest period.

**Supplementary Materials:** The following are available online at http://www.mdpi.com/1999-4907/11/11/1227/s1, Figure S1: Precipitation (mm) was recorded at a 2 m tower in an open space approximately 200 m from the studied site in 2017, Figure S2: Specific root length (SRL, m g-1) for different soil depths of Cryptomeria japonica and Chamaecyparis obtusa intact tree and cut tree plots.

**Author Contributions:** Conceptualization, M.F.; Methodology, M.F., A.K. (Ayumi Katayama), and T.H; Investigation, M.F., K.M., and S.J.; Data curation, M.F., S.M.M.S., and A.K. (Atsushi Kume); Writing—original draft preparation, M.F.; Writing—review and editing, M.F., M.C., A.K. (Ayumi Katayama), T.H., S.J., S.M.M.S., K.O., and A.K. (Ayumi Katayama); Project administration, K.O., and A.K. (Atsushi Kume); Funding acquisition, K.O., and A.K. (Atsushi Kume). All authors have read and agreed to the published version of the manuscript.

**Funding:** This study was partly supported by JSPS KAKENHI, grant numbers JP26292088 and JP18H04152.

**Acknowledgments:** The authors wish to gratefully acknowledge the staff of Kyushu University Forest for helping and supporting our laboratory and field experiments. We gratefully acknowledge Fukiko Kubota from the department of applied chemistry for helping and supporting our experiments in their laboratory with atomic absorption spectrometry Shimadzu AA-7000.

**Conflicts of Interest:** No potential conflict of interest was reported by the authors.

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
