# Peer review of "Relationship between Very Fine Root Distribution and Soil Water Content in Pre- and Post-Harvest Areas of Two Coniferous Tree Species"

_forests, doi:10.3390/f11111227_

Round 1
Reviewer 1 Report
Relationship between very fine root distribution and soil water content in pre and post-harvest areas of two coniferous tree species
After several careful readings, I can give my opinion:
- The title of the manuscript does not completely correspond to the content
- The majority of the data are previously published; what authors have clearly stated (Farahnak et al). Yet, in this manuscript only Root length density and Specific root length are unpublished data. Since mentioned data are given as a result of simple calculation procedure from previously publish data, I do not believe that the amount data are enough for publishing in Journal of this category
- Soil water content relationships with root characteristics is elaborated with soil properties (Ca, organic matter) yet nothing has been said about this parameter in introduction or material and method section. As a consequence, there is no clear connection between the manuscript sections
- The above also applies to repellency.
Overall, the study is interesting yet hard to read due to the poor presentation and continuity between the manuscript sections. Reader could easily get distracted by the fact that most of the data are previously published and that the majority of the discussion section is about the parameters that are poorly presented in introduction and material and method section.
I hope I have managed to give you clear and objective opinion
Best regards,
Author Response
We want to express our deep gratitude for giving us the opportunity to revise our manuscript. In this revised manuscript, lines were changed. Therefore, all responses are based on new changes. Changes in the manuscript were made with different coloring for reviewers (Blue: reviewer #1, Yellow: Reviewer #2, Green: Overlapped Reviewer #1 and #2).
Q1: The title of the manuscript does not completely correspond to the content.
Response: Thank you very much for your honest opinion. We made several changes in the introduction, materials and methods, and discussion. We believe, based on these changes, the title corresponds to the content.
Q2: The majority of the data are previously published; what authors have clearly stated (Farahnak et al). Yet, in this manuscript only Root length density and Specific root length are unpublished data. Since mentioned data are given as a result of simple calculation procedure from previously publish data, I do not believe that the amount data are enough for publishing in Journal of this category.
Response: Thank you very much for your consideration. In previous research (Farahnak et al. 2019a, b), we discussed soil hydraulic properties on the local scales (i.e., on the upslope and downslope of trees and stumps). This research used some soil properties from our previous publications (soil organic matter, soil hydraulic conductivity, and soil water repellency). We pool this data to explain more detailed fine root (biomass and length) and soil water content distribution in plot scales. Please see L107-L112 and L148-L149.
Q3: Soil water content relationships with root characteristics is elaborated with soil properties (Ca, organic matter) yet nothing has been said about this parameter in introduction or material and method section. As a consequence, there is no clear connection between the manuscript sections. The above also applies to repellency.
Response: Thank you very much for your attention to this matter. We added more explanation about soil nutrients availability impact on root distribution in the introduction (L43-L44). We also described these soil properties measurement methods in Materials and Methods (L117, L119-L120, L123-L128).
Q4: Overall, the study is interesting yet hard to read due to the poor presentation and continuity between the manuscript sections. Reader could easily get distracted by the fact that most of the data are previously published and that the majority of the discussion section is about the parameters that are poorly presented in introduction and material and method section.
Response: Thank you very much for your valuable time to read our manuscript and gave us your honest opinion. We improved our presentation quality and the continuity between the manuscript section by adding more detail about soil properties relationship with roots and soil water content vertical distribution in the introduction and the measurement methods. We believe that the current manuscript focuses more on the presented data in this research. The re-used data from previous research converted from local scale (previous publications) into the plot scale (current study) to explain root and soil water vertical distributions in pre and post-harvest areas.
Reviewer 2 Report
Please see attached file.

Author Response
We want to express our deep gratitude for giving us the opportunity to revise our manuscript. In this revised manuscript, lines were changed. Therefore, all responses are based on new changes. Changes in the manuscript were made with different coloring for reviewers (Blue: reviewer #1, Yellow: Reviewer #2, Green: Overlapped Reviewer #1 and #2).
Reviewer #2
The authors explored the root distribution in varying soil depths and how these are related to soil moisture. While the study uses robust experimentation and measurement protocols, the authors failed to use the maximum utility of their data (e.g. leaf litter, soil properties, climate data if there are more) but instead focus on root dynamics and soil moisture alone. Another weakness is the use of variables that were not measured in their study to explain the root distribution changes. Although these are necessary to explain mechanisms, let not this be the main reason for your measured variables' changes. Using unmeasured parameters will weaken your claims. I suggest making use of what is available and dig deeper into it. It also mentions the comparison of pre-and post-harvest coniferous trees, but the impact of the message they want to convey to the reader is a little weak. The graphical presentations as well make it hard to understand clearly the comparisons. The comparison can be strengthened by making effective figures backed up by thorough discussion into mechanisms and critical drivers of change. If all of these will be addressed, the manuscript will significantly be improved. I have here some comments and suggestions to help improve the manuscript.
Response: We appreciate your valuable time reviewing our manuscript and gave us your positive comments and the opportunity to polish our manuscript. Based on your comments and suggestions, we revised our manuscript as follows:
Lines (before revision) |
Comments |
Answers |
Lines (after revision) |
26 |
‘was not changed’ how about ‘did not change’? |
Thank you for your suggestion. “was not changed” changed to “did not change” |
26 |
28 |
‘suggested’ into ‘suggest’ |
Thank you for your suggestion. “suggested” changed into “suggest” |
28 |
164 |
What does ‘x’ means inside the box? Care to put a legend for easy reference of your boxes color? What does the error bar represents? |
Thank you very much for your attention. A legend for box plot information added to Figure 1 |
167 Figure 1 |
Figure 4 |
The figure header must be on the same page as the plots. Do not separate them in a page. Why correlation rather than regression models were used in this analysis? In regression models, you can evaluate the variance explained by every variable in every root category at varying soil depths. |
Thank you very much for your suggestion. We revised Figure 4 and put the header on the same page as the plots. We used the correlation between soil water content and very fine roots (biomass and length) to show that increasing density of very fine root (either biomass or length) has a negative impact on soil water content. This correlation means that in a slope with two different vegetation covers increasing very fine root density related to soil water content. When we remove the vegetation, decreasing in very fine root or very fine root decomposition (either biomass or length) causes water to deliver deeper soil (i.e., from 0-5 cm into the 5-30 cm). Certainly, many other variables impact soil water distribution but decomposed very fine root could be the dominant factor in Ch. obtusa cut tree plot. |
Figure 4 |
222 |
Fine root biomass of what species? |
Thank you very much for your careful question. Sorry for the unclear sentence. We meant fine root of both tree species. |
220-221 |
227 |
What do you ‘large very fine roots’ aren’t ‘large’ and ‘fine’ are opposite words? |
Thank you very much for your thoughtful question. We changed this phrase into “dense very fine root…” |
225-226 |
244-246 |
‘Archer et al. [49] showed that the larger root length in an 244 Anthyllis site, when compared with a Reatama site, could be because of the ability of soil in the 245 Anthyllis site to store a larger amount of water that is released at low suction’. What is the relevance of this sentence into your discussion in this paragraph 221- 257. |
Thank you very much for your attention. We removed this sentence from the discussion. |
|
258-260 |
Be careful to always use Ca2+ in many of your explanations as you did not measure this? These remain speculations and assumptions. Maybe focus on what data you got, the relationships of your dependent and independent variables. You have a wealth of data to explore more. Have confidence on what you have got. |
Thank you very much for your kind suggestion and opinion. We added a measurement method of Ca2+, and data for Ca2+ was already available in Table 1. Therefore, it makes sense now to explain the relationship between Ca2+ and vertical root distribution. |
123-126 |
265-267 |
Vague sentence. |
Thank you very much for your attention. A clearer sentence was added. Here, we tried to explain that despite differences between intact tree and cut tree plots sampling time, the soil water content of the intact tree and cut tree plots of each tree species show a similar trend by increasing soil depth. For example, the soil water content of Cr. japonica intact tree plot and cut tree plots show a similar decreasing trend by increasing soil depths. Between Ch. obtusa intact tree and cut tree plots also similar soil water content observed (but opposite from Cr. japonica) that increased by increasing soil depths. |
262-268 |
267-268 |
The previous sentence did not say anything like this |
Thank you very much for your consideration. Based on changes in the previous sentence and this sentence. Now there is a connection between two sentences. |
258-264 |
268-272 |
At which result in the figure do you refer to this? Perhaps make a regression about these relationships to back up your claim |
Thank you very much for your attention. This result is related to Figure 1 and Figure A1. Figure A1 shows the sampling time of intact tree plots is different from cut tree plots. Figure 1 shows the soil water content similarity between intact tree plots and cut tree plots by increasing soil depths (a decreasing trend in Cr. japonica plots and an increasing trend at Ch. obtusa plots) |
262-268 |
274-275 |
‘Additionally, by removing trees, annual 274 plants had opportunities to take up water (Figure 3)’ Which part in the figure does this presented? |
Thank you very much for your careful attention. This refers to denser other root length at Ch. obtusa cut tree plot than intact tree plot. In figure 3, other root length part. |
192-198 Figure 3 (other root length) |
Boxplots Figures |
The figures are confusing to see, especially at first glance. I suggest if you can plot it clearly, so much the better. Below are my hypothetical example such that the reader may not be confused on what to see in the figure. You can do either of these or any graphical presentation you like in excel or R or any software, without altering your results. I just want to see your figures aesthetically prepared if possible. You can add error bars and letters for significant differences. If you cannot make like this, at least make your graphs very easy to understand. |
Thank you very much for your kind suggestion. We made some new Figures based on your incredible recommendations. |
Figure 2-5 and Figure A2 |
277-278 |
How about the concept of hydraulic lift and hydraulic redistribution? Since the tree is severed by cutting, there is lesser way for the trees to promote hydraulic lift and redistribution, causing the water at 5-30 cm level stagnant, especially if drainage flow is low. The soil water is left untapped at 5-30 cm since the vegetation is only annual plants that are shallow-rooted and less consumer of soil water. Most likely, the soil water at 0-5cm is more likely to be evaporated as it is exposed or wasted as surface run-off. |
Thank you very much for your opinion about hydraulic lift and hydraulic redistribution. In this study, we concentrated on 0-30 cm soil depth. In soil science, this depth is considered shallow soil whereas hydraulic lift or redistribution covers a much wider soil profile range. That means lifting water from groundwater and use it for transpiration (daytime) and redistributed in soil (nighttime). Therefore, in the case of our study, we stressed very fine root existence at 0-5 cm in Ch. obtusa intact tree plot and soil water movement from 0-5 to 5-30 soil depths in Ch. obtusa cut tree plot. |
|
278-282 |
This sentence is vague. What does it mean here? That the soil moisture in cut trees in japonica is greater than in intact tree plots? Unless there is no proper drainage, the soil moisture must have been stagnant at these sites? Without a tree canopy, evapotranspiration is expected to rise due to exposure to direct sunlight. |
Thank you very much for your thoughtful questions. Cr. japonica cut tree plot is a thinned plot. Therefore, canopy cover is still existing. In addition to this, the soil surface is covered with a thick litter layer. We suggested that these conditions caused higher soil surface water content (at 0-5 cm) in Cr. japonica cut tree plot than the intact tree plot. |
L271-L272 |
289-294 |
Do not draw a conclusion in your study based on something you do not measure e.g. fine root decomposition. |
Thank you very much for your consideration. Because very fine root (biomass and length) decreased significantly after harvesting Ch. obtusa trees, we concluded that decomposed roots could be a pathway for water redistribution in cutting areas (Ch. Obtusa cut plot). |
L280-L290 |
|
Please correct some grammatical errors. Have your manuscript be proof-read by a native English speaker. |
Thank you very much for your kind consideration. We double-checked our manuscript and corrected any grammatical errors (yellow highlighted). Our manuscript also proof-read by an English company (Edanz). |
299-304 |
Round 2
Reviewer 1 Report
Dear Authors, thank you for your effort to improve the manuscript yet my opinion stays the same because of the previously given reasons; the poor connection between the title-results-discussion sections and previously published study results.
Author Response
We want to thank you for your valuable time in reviewing our manuscript. The graphs redesign by adding different colors for the readers' better visibility (Fig 2, 3, 4 and A1, and A2). We improved our graphical abstract to guide potential readers through the content of the manuscript. We also added Fig 5 in the discussion part to show the overall summary of discussion and this study (L220-L225). We believe Fig 5 provides enough information for the readers to follow this study's story as they go through discussion.
Reviewer 2 Report
Thank you for addressing each and every comment I made, especially on your graphs. You did it excellently!
Author Response
We want to thank you for your valuable time in reviewing our manuscript. We appreciate your great suggestions to improve our manuscript. The graphs redesign by adding different colors for the readers' better visibility (Fig 2, 3, 4 and A1, and A2). We improved our graphical abstract to guide potential readers through the content of the manuscript. We also added Fig 5 in the discussion part to show the overall summary of discussion and this study (L220-L225). We believe Fig 5 provides enough information for the readers to follow this study's story as they go through discussion.